# Effects of Soy-Based Infant Formula on Weight Gain and Neurodevelopment in an Autism Mouse Model

**DOI:** 10.3390/cells11081350

**Published:** 2022-04-15

**Authors:** Cara J. Westmark, Mikolaj J. Filon, Patricia Maina, Lauren I. Steinberg, Chrysanthy Ikonomidou, Pamela R. Westmark

**Affiliations:** 1Department of Neurology, University of Wisconsin, Madison, WI 53706, USA; filon@wisc.edu (M.J.F.); mainapatricia@gmail.com (P.M.); lsteinberg@luc.edu (L.I.S.); ikonomidou@neurology.wisc.edu (C.I.); prwestmark@wisc.edu (P.R.W.); 2Molecular Environmental Toxicology Center, University of Wisconsin, Madison, WI 53706, USA; 3Undergraduate Research Program, University of Wisconsin, Madison, WI 53706, USA; 4Molecular Environmental Toxicology Summer Research Opportunities Program, University of Wisconsin, Madison, WI 53706, USA

**Keywords:** fragile X, *Fmr1^KO^*, obesity, soy-based infant formula

## Abstract

Mice fed soy-based diets exhibit increased weight gain compared to mice fed casein-based diets, and the effects are more pronounced in a model of fragile X syndrome (FXS; *Fmr1^KO^*). FXS is a neurodevelopmental disability characterized by intellectual impairment, seizures, autistic behavior, anxiety, and obesity. Here, we analyzed body weight as a function of mouse age, diet, and genotype to determine the effect of diet (soy, casein, and grain-based) on weight gain. We also assessed plasma protein biomarker expression and behavior in response to diet. Juvenile *Fmr1^KO^* mice fed a soy protein-based rodent chow throughout gestation and postnatal development exhibit increased weight gain compared to mice fed a casein-based purified ingredient diet or grain-based, low phytoestrogen chow. Adolescent and adult *Fmr1^KO^* mice fed a soy-based infant formula diet exhibited increased weight gain compared to reference diets. Increased body mass was due to increased lean mass. Wild-type male mice fed soy-based infant formula exhibited increased learning in a passive avoidance paradigm, and *Fmr1^KO^* male mice had a deficit in nest building. Thus, at the systems level, consumption of soy-based diets increases weight gain and affects behavior. At the molecular level, a soy-based infant formula diet was associated with altered expression of numerous plasma proteins, including the adipose hormone leptin and the β-amyloid degrading enzyme neprilysin. In conclusion, single-source, soy-based diets may contribute to the development of obesity and the exacerbation of neurological phenotypes in developmental disabilities, such as FXS.

## 1. Introduction

Obesity is an epidemic in the United States where 19% of children, 21% of adolescents, and 42% of adults have a body mass index (BMI) greater than the 95th percentile [1,2]. In addition, 8% of infants and toddlers have high weight-for-recumbent length measurements, putting them at risk for developing childhood obesity [3]. Obese children are more likely to exhibit attention deficit/hyperactivity disorder (ADHD), depression, learning disability, and developmental delay [4]. Neurological phenotypes can be exacerbated by metabolic conditions. For example, diabetes, hypertension, and obesity during pregnancy are associated with autism spectrum disorders (ASD) and developmental delays in the offspring [5]. Contemporary hypotheses propose that exposure to environmental or dietary chemicals during gestation and/or postnatal development may predispose infants to developing obesity, ADHD, and/or autism [6,7,8]. A critical gap in the literature is the identification of specific foods or food-specific chemicals associated with body weight and neurodevelopmental phenotypes.

Serendipitous observations in our laboratory suggest that the consumption of soy-based diets is associated with weight gain in *Fmr1^KO^* mice, a mouse model utilized to study fragile X syndrome (FXS) and autism. FXS is the most common form of inherited intellectual disability and the leading known genetic cause of autism. The major clinical phenotypes associated with this X-chromosome-linked disorder are highly variable intellectual disability (overall IQ < 70), autistic-like behaviors, seizures, macrocephaly, and macroorchidism [9]. In most cases, FXS is caused by a trinucleotide repeat expansion mutation in the 5′-UTR of the *FMR1* gene [10] that is associated with transcriptional silencing of the *FMR1* promoter and loss of expression of fragile X messenger ribonucleoprotein (FMRP) [11]. FMRP is a pivotal mRNA-binding protein with roles in translational regulation, genome stability, RNA editing, mRNA target binding, microRNA regulation, cell differentiation, ion channel binding, calcium signaling, excitatory/inhibitory balance, and activity-dependent synapse development [12]. FMRP expression is absent or greatly reduced in FXS, and many FXS phenotypes are manifested in *Fmr1^KO^* mice, which lack expression of FMRP [13]. FXS research to date has primarily focused on the brain and the study of severe cognitive and behavioral deficits; however, there is also a significant weight problem associated with the disorder. Children with FXS show accelerated prepubescent growth, but a diminution of the normal pubertal growth spurt [14,15]. Male children with FXS have higher rates of obesity (31%), compared to their typically developing same-aged peers (18%) [16].

During preclinical testing to evaluate the efficacy of anti-convulsant drugs, we observed that diet (soy-based chow versus casein-based purified ingredient diet) significantly affects seizure propensity and weight gain in juvenile *Fmr1^KO^* mice [17]. The human correlate of juvenile mice maintained on a single-source soy-based rodent chow is babies exclusively fed soy-based infant formula. We searched the evidence-based literature regarding possible adverse health effects associated with consumption of soy-based infant formulas but found a dearth of published reports and no studies specific to developmental disorders such as FXS. To address this gap in the literature, we conducted retrospective medical record and survey analyses and found associations between postnatal consumption of soy-based infant formula and an increased incidence of seizures and/or autistic behavior in autism and FXS [18,19,20,21]. Here, we hypothesize that the high consumption of soy protein during postnatal development is a dietary exposure that increases the risk of developing obesity, particularly in vulnerable populations such as FXS.

Herein, we conducted secondary analyses of our laboratory seizure records that contained body weight metrics as a function of diet and mouse age and genotype. We also conducted a prospective evaluation of weight gain, blood plasma biomarkers, and behavior as a function of consumption of soy-based infant formula diets in WT and *Fmr1^KO^* mice. Finally, we conducted a secondary analysis of infant growth metrics as a function of postnatal diet (breast milk, soy-based infant formula, cow milk-based infant formula) using the Centers for Disease Control and Prevention (CDC) Infant Feeding Practices Study II (IFPSII) and Year 6 Follow-Up (Y6FU) datasets. These studies are necessary as controversy surrounds the effects of soy on metabolic phenotypes and little is known regarding the molecular signature of soy consumption, particularly in disease populations. We found that soy-based diets increased weight gain in *Fmr1^KO^* mice irrespective of age. A soy-based infant formula diet altered the expression of several mouse plasma proteins, including the adipose hormone leptin and the β-amyloid degrading enzyme neprilysin, as well as affected learning and repetitive behaviors. Consumption of soy-based infant formula was not associated with increased obesity in typically developing human infants followed out to 6 years of age, but was associated with an increased need for support in school, respiratory allergies, and ADD/ADHD.

## 2. Materials and Methods

Mouse Husbandry: Mice were obtained from our colony, which has been maintained at the University of Wisconsin-Madison for over 15 years; the strain was originally provided by the laboratory of Dr. William Greenough (University of Illinois, Urbana-Champaign, IL, USA). WT and *Fmr1^KO^* mice are in the C57BL/6J background, the most widely used inbred mouse strain, which has been extensively utilized for FXS and obesity research [13]. Breeding pairs were housed in microisolator cages on a 12 h (0600–1800) light cycle with ad libitum access to food and water (specific diets are described below). Genotypes were determined by PCR analysis of DNA extracted from tail biopsies taken at weaning [22].

Diet Formulations: Purina 5015 mouse diet is a complete life-cycle diet designed to support reproduction, growth, and maintenance of mice (LabDiet, St. Louis, MO, USA). This high-energy chow supports maximum reproduction for postpartum breeding for females under the simultaneous stress of lactation and gestation. The main ingredients include ground wheat, dehulled soybean meal, and ground corn. Calories are provided in the form of 18% protein, 26% fat, and 56% carbohydrate. Teklad 2019 is a fixed formula, extruded diet also designed to support the gestation, lactation, and growth of rodents, but does not contain any alfalfa, soybean meal, animal protein, or fish meal (Envigo, Fitchburg, WI, USA). The absence of alfalfa and soybean meal minimizes the occurrence of natural phytoestrogens to less than 20 mg/kg, as opposed to 120–340 mg/kg in Purina 5015 [23]. The absence of animal protein and fish meal minimizes the presence of nitrosamines. The main ingredients include ground wheat, ground corn, corn gluten meal, and wheat middlings. Calories are provided in the form of 23% protein, 22% fat, and 55% carbohydrates. D07030301 is a purified ingredient diet based on the AIN-76A formulation but modified to match the macronutrient content of Purina 5015 (Research Diets, Inc., New Brunswick, NJ, USA, Appendix A) [17]. The main ingredients are corn starch, casein, and maltodextrin-10. Purified ingredient diets are phytoestrogen-free. AIN-93G is a newer formulation of the purified ingredient diet recommended by the American Institute of Nutrition for the growth, pregnancy, and lactation of rodents [24], with the main ingredients of corn starch, casein, and maltodextrin-10, and calories provided in the form of 20% protein, 16% fat, and 64% carbohydrates (Research Diets, Inc., New Brunswick, NJ, USA). Powdered forms of Enfamil Premium (cow milk-based) and ProSobee (soy protein-based) human infant formulas were purchased from Wal-Mart (www.walmart.com; last accessed on 13 April 2022). Food pellets were produced from the powdered infant formulas (Research Diets, Inc., New Brunswick, NJ, USA) (Appendix A). Many micronutrients in the infant formulas exceeded National Research Council (NRC) recommended levels for mice; those that were deficient were supplemented including DL-methionine, choline, and some vitamins and minerals (Appendix A). The main ingredients in Enfamil Premium include nonfat milk, lactose, vegetable oil (palm olein, coconut, soy, and high-oleic sunflower oils) and whey protein concentrate. The main ingredients in Enfamil ProSobee are corn syrup solids, vegetable oil (palm olein, coconut, soy, and high-oleic sunflower oils), and soy protein isolate. Calories (kcal %) are provided as 8% protein, 47% fat, and 44% carbohydrates for the cow milk-based infant formula diet (CIF; D14010402 based on Enfamil Premium) and as 10% protein, 48% fat, and 42% carbohydrates for the soy-based infant formula diet (SIF; D14010401 based on Enfamil ProSobee). Thus, CIF and SIF contain significantly less protein and more fat than typical mouse diets. Soy oil, which is present in both CIF and SIF, does not contain phytoestrogens. All test diets provide about half of their energy, 42–64%, through carbohydrates.

Retrospective Analysis of Mouse Seizure Datasets: As part of our audiogenic-induced seizure (AGS) testing protocol in mice, body weight data on juvenile animals (postnatal day 21, P21) was collected over a 5-year period. Herein, those data were analyzed by 2-way ANOVA with post hoc Tukey’s multiple comparison test to assess differences in body weight dependent on *Fmr1* genotype and diet. As part of a protocol to test pentylenetetrazol-induced seizure propensity in adult wild type (WT) mice, body weight data was collected on 2-month-old mice. Herein, those data were analyzed by Student’s *t*-test. As part of the prospective testing described below, adult male *Fmr1^KO^* mice were randomized to D07030301, CIF or SIF at 2–3 months of age for breeding purposes and weighed at approximately 4 months of age; the data were analyzed by 1-way ANOVA with post hoc Tukey’s multiple comparison tests.

Prospective Mouse Diet Studies: Study 1: Mice were conceived and maintained on Teklad 2019 diet, weaned, and weighed at P21, and maintained on Teklad 2019 until P30, at which time the males were randomized to Teklad 2019, CIF, or SIF. Mice were weighed every 3 days from P18 to P30 and once per week thereafter. A reference cohort included mice bred and maintained on AIN-93G throughout the study. Mice were anesthetized with isoflurane before blood was collected from the inferior vena cava with a 21-gauge butterfly needle, mixed with sodium heparin anticoagulant, spun at 5000 rpm for 10 min, and the plasma layer quick frozen and stored at −80 °C. Blood plasma was shipped to RayBiotech, Inc. and used to probe mouse protein arrays (RayBiotech, Inc., Norcross, GA, USA; catalog #QAM-CAA-4000; 200 target proteins). Replicates (*n* = 3) were averaged and presented as means and standard deviations of the mean. Two-way ANOVA was used to determine statistical significance. Study 2: Early postnatal exposure to CIF and SIF was tested by randomly assigning breeding pairs to CIF or SIF diets with continuous maintenance on their respective diets throughout pregnancy and lactation. Pups were weaned and weighed at P21. Study 3: Mice were conceived and maintained on Teklad 2019 diet; males and females were weaned, weighed, and randomized to Teklad 2019, CIF, and SIF diets at P20; weighed weekly for 2 months; tested for normal mouse activity (nest building, P47) [25], learning and memory (passive avoidance, P51) [26], and body composition (EchoMRI, P62 and P76).

Mouse Behavior: Nest building was conducted as described [25]. Briefly, corncob bedding and two 2X2″ white cotton nestlets (catalog #NES3600; Ancare, Baltimore, NY, USA) were added to clean standard cages with feed and water. A single mouse was transferred to a nestlet cage at 3 p.m. At 7 a.m. the following day, the mouse was returned to its home cage. The empty nestlet cage was photographed and the length, width, and height of the nests were measured. Nests were scored based on the following scale: (0) nestlet intact, (1) flat nest with partially shredded material, (2) shallow nest with shredded material but lacks fully formed walls, (3) nest with well-developed walls, and (4) nest in shape of cocoon with partial or complete roof. The volume of the nest was calculated by multiplying the length x height x width of the nest. Average data was plotted ±SEM and statistical significance determined by 2-way ANOVA with post hoc Tukey’s multiple comparison tests.

Passive avoidance was conducted as described [26]. Briefly, mice were acclimated to the experimental room for at least 20 min prior to testing in a foot shock passive avoidance paradigm using an aversive stimulator/scrambler (Med Associates Inc., St. Albans, VT, USA). A bench-top lamp was turned on behind the center of a light/dark shuttle box and aimed toward the back-left corner away from the dark side of the shuttle box. On the training day, a mouse was placed in the light side of the shuttle box toward the back corner away from the opening to the dark side of the shuttle box. The trap door in the shuttle box was open. After the mouse crossed over to the dark side, the trap door was closed and the latency time for the mouse to move from the light to the dark side was recorded. The mouse was allowed to equilibrate in the dark side for 5 seconds before receiving a 2-second 0.5 mA foot shock. After 15 seconds, the mouse was removed from the shuttle box and returned to its home cage. The apparatus was cleaned with 70% EtOH between animals. At test times (6, 24 and 48 hours after training), the mouse was placed in the light side of the shuttle box facing the left rear corner away from the opening to the dark side with the trap door open. After the mouse crossed to the dark side, the trap door was closed and the latency time for the mouse to move from the light to the dark side was recorded. If the mouse did not move to the dark side within 300 seconds, it was gently guided to the dark side and the trap door was closed. The mouse was allowed to equilibrate to the dark side for 5 seconds before return to the home cage. Mice only received one shock on the training day. Testing at 24 and 48 hours measured extinction. Average data was plotted ±SEM and statistical significance determined by 2-way ANOVA with post hoc Tukey’s multiple comparison tests.

EchoMRI: Awake mice were placed in an enclosed tube and inserted into an EchoMRI instrument (EchoMRI LLC, Houston, TX, USA) where nuclear magnetic resonance (NMR) was used to quantify total body weight, lean mass, fat mass, and water mass in grams. Average data was plotted ±SEM with statistical significance determined by 2-way ANOVA with post hoc Tukey’s multiple comparison tests.

Analysis of CDC Infant Feeding Practices II Study (IFPSII) and Year 6 Follow-Up (F6FU) Datasets: The Food and Drug Administration (FDA) and CDC in collaboration with NIH and the USDA conducted a longitudinal consumer-based research study entitled, “Infant Feeding Practices II”, with the goal of understanding and improving the health status of mothers and children. A series of monthly questionnaires were used to collect information from the mothers from the 7th month of pregnancy through the infant’s first year of life. Specifically, the study collected detailed information on the following: (1) foods fed to the infants including breast milk and infant formula, (2) factors that may contribute to infant feeding practices and breastfeeding success, (3) mother’s intrapartum hospital experiences, sources of support, and postpartum depression, (4) mother’s employment status and child care arrangements, (5) infant sleeping arrangements, (6) other issues such as food allergies, experiences with breast pumps, and WIC participation, and (7) diets of pregnant and postpartum women. Four questionnaires (prenatal, maternal dietary intake, birth screener, and neonatal) as well as nine modules were completed and the deidentified data made publicly available by the CDC. Participants in the IFPSII included approximately 4000 pregnant women from across the United States whose infants were born between May of 2005 and March of 2006. The sample was selected from a national consumer opinion panel consisting of 500,000 households. Mothers (2250 women) qualified and continued their participation through the baby’s first year. To qualify, a healthy woman must have given birth to one healthy, full-term, or near-term infant weighing at least 2.5 kg at birth. The study population was oversampled for low-educated, African American, and Hispanic women, as well as women living in the Breastfeeding Awareness Campaign’s Community Demonstration Project areas to increase representation from these groups. There was a Year 6 Follow-Up study (Y6FU) of children participating in the IFPSII that consisted of a single questionnaire administered in 2012 and resulting in 1542 completed questionnaires that included information on development and comorbid disorders [27]. The preliminary analyses examined long-term child health outcomes at 6 years of age [28]. The IFPSII and Y6FU were approved by the Research Involving Human Subjects Committee of the Food and Drug Administration. Datasets were provided as Statistical Analysis Software (SAS) files from the CDC.

The IFSII and Y6FU datasets were merged to obtain infant feeding, growth data, and developmental outcomes for the 1542 subjects participating in both surveys. Variables of interest from the IFSII included the following: sex (N1); exclusive breastfeeding (nexbf, M2EXBF and M3EXBF); use of cow milk formula for 3 months without any reported use of soy-based infant formula, and use of soy-based infant formula for 3 months without any reported use of cow milk formula (N47-2, N47-3, M2A10-2, M2A10-3, M3A10-2, M3A10-3, N47-4, N47-5, M2A10-4, M2A10-5, M3A10-4, M32A10-5), weight (M3A28, M5A32, M7A31, MWA34), and height (M3A30IN, M5A34IN, M7A33IN, MWA36IN). Variables of interest from the Y6FU included the following questions from Section A: Q5. During this school year, has a special plan been developed at school to provide your 6-year-old with extra help or support such as a special needs program or an individualized education program (IEP)? Q6. During this school year, has your 6-year-old received any of the following services? Speech or language therapy; Occupational therapy or other type of therapy for help with handwriting or other motor skills; Special instruction or help in one or more school subjects such as reading or math; Special services because of a problem with vision or hearing; Psychological services or counseling because of a problem with emotions, behavior, or socialization; Behavioral support, such as a behavior management plan or individual support in the classroom by an assistant; Other; None of these. Questions from Section B included the following: Q1. How tall is your 6-year-old now (without shoes)? (Tape measure enclosed). Q2. How much does your 6-year-old weigh now (without shoes)? Q22A. Has a doctor or other health professional ever told you that your 6-year-old has any of the following conditions? (a) Hearing problems, (b) A digestive problem such as colitis, acid reflux, colic, or Crohn’s, (c) Eczema or any kind of kind of skin allergy (such as contact dermatitis), (d) Hay fever or respiratory allergy (to pets, pollen, mold, dust mites, etc.), (e) Asthma, (f) Diabetes, (g) Attention Deficit Disorder or Attention Deficit Hyperactivity Disorder, ADD or ADHD, (h) Autism or developmental delay, (i) Depression or anxiety.

Subjects were binned by sex, exclusive breast milk use for 3 months, and use of cow milk or soy-based infant formula for 3 months. BMI was calculated as (weight)/(height^2^) in metric units. Weight, height, and BMI data at 3, 5, 7, 12, and 72 months of age were analyzed by mixed-effects ANOVA with matching by age, and grouped data plotted as min-to-max box plots using Prism version 9.2.0 (GraphPad Software, LLC, San Diego, CA, USA). Categorial data were analyzed by Chi square, or Fisher exact test if there were less than 5 subjects in any row or column variable. Statistical significance was denoted by *p* < 0.05.

## 3. Results

### 3.1. Soy/Grain-Based Rodent Diet Is Associated with Increased Weight Gain in Juvenile Fmr1^KO^ Mice

During the study of seizure susceptibility in mice [17], we observed that juvenile *Fmr1^KO^* mice fed soy protein/grain-based Purina 5015 lab chow exhibited increased weight gain (22%) compared to *Fmr1^KO^* mice maintained on a casein-based, purified ingredient diet D07030301 (Figure 1A). D07030301 is a modified version of the American Institute of Nutrition 76A (AIN76A) diet that was matched to Purina 5015 for protein, fat, and carbohydrate content (Appendix A). Mice were weighed at postnatal day 21 (P21) immediately prior to seizure testing. A comparison of four diets, including Purina 5015 and D07030301 plus an additional chow (Teklad 2019) and purified ingredient diet (AIN-93G), indicated a statistically significant increase in body weight in WT mice fed Purina 5015 compared to AIN-93G and Teklad 2019 but not D07030301 (Figure 1A). D07030301 elicited increased weight gain in WT mice in comparison to Teklad 2019. The Purina 5015 and D07030301 have somewhat lower protein and higher fat energy densities than the AIN-93G and Teklad 2019. AIN-93G has a significantly lower fat but higher carbohydrate energy density. In contrast, weight gain in *Fmr1^KO^* mice was significantly higher in response to the Purina 5015 and equivalent among the 3 low/no phytoestrogen diets irrespective of fat and protein energy densities. WT female mice maintained on D07030301 weighed more than *Fmr1^KO^* females on the same diet, while *Fmr1^KO^* females maintained on Purina 5015 weighed more than those on D07030301 (Appendix A). There were no significant differences in body weight dependent on diet in Tg2576 mice, a model for Alzheimer’s disease, for males fed Purina 5015 (9.37 ± 0.43, *n* = 10) compared to D07030301 (8.82 ± 0.17, *n* = 43; Student’s t-test *p* = 0.19), although the results approached statistical significance for a 12% increase in body weight in female Tg2576 fed Purina 5015 (9.30 g ± 0.42, *n* = 11) compared to D07030301 (8.27 ± 0.25, *n* = 43; Student’s *t*-test *p* = 0.07) (data not shown).

Body weight data generated by Jackson Laboratories for C57BL/6J mice (catalog #000664) at 3 weeks of age indicate that males weigh 10.6 g (SD 1.9) and females weigh 10.1 g (SD 1.7) (https://www.jax.org/jax-mice-and-services/strain-data-sheet-pages/body-weight-chart-000664; last accessed on 13 April 2022), and is based on mice maintained on a grain-based chow containing not less than 6% fat and 18% protein (LabDiet^®^ 5K52/5K67). The major ingredients include ground wheat, ground corn, wheat middlings, ground oats, fish meal, and dehulled soybean meal. The ingredients are most similar to those in Purina 5015. The energy content is most similar to that of AIN-93G. *Fmr1^KO^* male mice maintained on AIN-93G and both WT and *Fmr1^KO^* male mice maintained on Teklad 2019 exhibited average body weights more than one standard deviation below the average body weight reported for C57BL/6J mice by Jackson Laboratories. Female mice were within one standard deviation of Jackson Laboratory values.

These data indicate that postnatal growth is not solely dependent on the energy density of the diet, as the Purina 5015 and D07030301 diets were matched for percent protein, fat, and carbohydrate. The data also suggest that the *Fmr1* genotype likely interacts with diet to affect juvenile weight gain in mice. The major difference between Purina 5015 and D07030301 was the protein source, soy, and grains versus casein, respectively. Ground wheat and corn are major ingredients in both the Purina 5015 and Teklad 2019 chows; however, only the Purina 5015 contains soybean meal. A major limitation to this retrospective analysis is that the data were collected over several years without a rigorous planned study design to evaluate the effect of diet on weight gain.

### 3.2. Purina 5015 Does Not Statistically Alter Weight Gain in Adult Animals

We also asked if the Purina 5015 and D07030301 diets affected weight gain in adult animals. During testing these diets on pentylenetetrazol (PTZ)-induced seizure susceptibility in adult animals, C57BL/6J mice were maintained on Purina 5015 for 1 month and then transferred to D07030301 or left on Purina 5015 for an additional 1 month prior to seizure testing (Appendix A). Differences in weight gain (4% increase in females and 5% increase in males fed Purina 5015) approached but did not reach statistical significance with a minimum of 20 mice per cohort (Student’s *t*-test, *p* = 0.07). These data suggest that fast growing, juvenile animals are more susceptible to the effects of diet on weight gain. We hypothesized that soy protein was contributing to seizure and weight phenotypes.

### 3.3. SIF Increases Weight Gain in Adolescent Fmr1^KO^ Mice Compared to CIF

The human correlate of mice fed single-source soy protein-based lab chow is babies fed soy-based infant formula. We tested the casein- and soy-based infant formulas on weight gain in WT and *Fmr1^KO^* mice. Powdered Enfamil Premium^®^ (cow milk-based infant formula) and Enfamil ProSobee^®^ (soy-based infant formula) were purchased from Wal-Mart and formulated into feeding pellets (CIF and SIF, respectively) (Appendix A). The protein energy densities of CIF and SIF were significantly lower than the mouse diets tested, 8% CIF and 10% SIF, compared to 18–23% in mouse diets. We chose to formulate powdered infant formula into feeding pellets, as opposed to adding soy protein isolate to a purified ingredient diet, to closely mimic what infants are fed.

Growth charts were generated for WT and *Fmr1^KO^* male mice in response to CIF and SIF over a 1-month period (Study 1). Mice were conceived and maintained on the Teklad 2019 diet until P30, at which time they were randomized to Teklad 2019, CIF or SIF (Figure 1B). A fourth cohort was conceived and maintained on AIN-93G throughout the study. There was no difference in weight gain based on genotype nor between the Teklad 2019, SIF, and AIN-93G diets at all time points tested (Figure 1C). These growth curves mirror the expected growth trajectories for C57BL/6J mice, as determined by Jackson Laboratories. There was a statistically significant decrease in body weight at all time points tested in *Fmr1^KO^* mice maintained on CIF compared to SIF. The two-way ANOVA did not indicate statistical significance in the WT mice as a function of diet, although the growth curves followed the same pattern as the *Fmr1^KO^* mice. Casein alone is not sufficient to reduce body weight as the AIN-93G and Teklad 2019 growth curves overlapped. Considering the similar energy densities of CIF and SIF and that the adolescent mice were still rapidly growing, these data support the conclusion that soy-based diets contribute to increased weight gain in mice. In this case, the growth curves of the SIF cohorts mirrored the low/no phytoestrogen mouse diets (Teklad 2019 and AIN-93G), suggesting that the suboptimal protein content of the SIF was compensated for by a bioactive component in or associated with soy protein.

### 3.4. CIF and SIF Are Suboptimal for Mouse Breeding

We assessed early postnatal treatment with CIF or SIF (Study 2); however, all male and female *Fmr1^KO^* pups maintained on either CIF or SIF throughout gestation exhibited more than one standard deviation decrease in expected body weight compared to Jackson Laboratories C57/BL6 mice (Appendix A). Although several litters were generated, the CIF and SIF diets were not optimal for rodent breeding or postnatal pup growth. The infant formulas contained adequate fat and carbohydrates for mouse breeding and lactation, but lower than recommended protein.

### 3.5. SIF Is Associated with Obesity in Adult Male Fmr1^KO^ Mice

Interestingly, the adult male *Fmr1^KO^* mice that were maintained on a SIF diet for breeding purposes developed obesity (Figure 1D). Mice were randomized to D07030301, CIF or SIF diets at 2–3 months of age for breeding. Mice in the SIF cohort became noticeably obese compared to those fed CIF or D07030301; hence, all mice were weighed at approximately 4 months of age. The D07030301 mice weighed an average of 28.4 g (SD 1.3) at 16.5 weeks of age (SD 0.95) compared to the expected Jackson Laboratory weight of 30.5 g (SD 0.5). The CIF mice weighed an average of 29.4 g (SD 2.6) at 17.5 weeks of age (SD 2.3) compared to the expected Jackson laboratory weight of 30.7 g (SD 1.1). The SIF mice weighed an average of 40.3 g (SD 9.3) at 18.1 weeks (SD 3.4) compared to the expected Jackson laboratory weight of 31.0 g (SD 1.4), resulting in a 37–42% increase in body weight compared to CIF and D07030302 (*p* < 0.02).

### 3.6. Consumption of SIF Is Associated with Altered Plasma Biomarker Expression

Biomarker expression was measured in WT and *Fmr1^KO^* mice after maintenance on Teklad 2019, CIF and SIF diets (Appendix A, Study 1 mice). Differentially expressed proteins are categorized based on genotype and diet (Table 1, Table 2 and Table 3). Of note, neprilysin (NEP) was the only target on the arrays that exhibited altered expression in response to SIF in both WT and *Fmr1^KO^* mice. Leptin was elevated in WT mice fed SIF.

### 3.7. SIF Is Associated with Altered Behavior

Next, we asked if SIF affected mouse behavior as a function of the *Fmr1* genotype (Study 3). First, we repeated the growth charts, albeit using both female (*Fmr1^HET^*, *Fmr1^KO^*) and male (WT, *Fmr1^KO^*) littermate mice in response to Teklad 2019, CIF and SIF diets. Pups were weaned at P20 and randomized to their respective diets (Figure 2A). There were no statistical differences in weight gain based on genotype nor between the Teklad 2019 and SIF diets at all time points tested, similar to Figure 1C. There was a statistically significant decrease in body weight in *Fmr1^KO^* females from P27-P48, and at all time points after treatment in WT and *Fmr1^KO^* males, maintained on CIF compared to SIF. The mixed effects, two-way ANOVA did not indicate statistical significance in the *Fmr1^HET^* female mice fed CIF versus SIF except at P34. The *Fmr1^HET^* and *Fmr1^KO^* CIF growth curves overlapped, but the growth curve for the *Fmr1^HET^* mice fed SIF indicated a trend for lower body weights at all time points compared to *Fmr1^KO^* fed SIF. Analysis of body composition by EchoMRI indicated increased average body weight in male WT and male and female *Fmr1^KO^* mice fed SIF versus CIF due to increased lean mass and not body fat at P62 and P76 (Figure 2B, Appendix A). Others show higher body weight and lean and fat mass by EchoMRI in 10-week-old *Fmr1^KO^* (FVB/129 background) mice fed a Teklad 18% protein diet [29], suggesting strain-specific differences. *Fmr1^KO^* males fed SIF exhibited a 56% decrease in nest building scores (Figure 2C). WT male mice fed SIF exhibited a 4.6-fold increase in learning (Figure 2D). Increased learning in WT male mice in response to SIF in the passive avoidance test concurs with elevated Children’s Memory Scale (CMS) Index scores in typically developing males in the Beginnings Study soy infant formula cohort [30].

### 3.8. Soy-Based Infant Formula Is Not Associated with Obesity in Typically Developing Subjects at 6 Years of Age

An increased infant BMI is associated with an obesity status at 5 years of age [31,32]. Based on the mouse data, we postulated that consumption of soy-based infant formula during postnatal development could be contributing to the childhood obesity epidemic. Using the IFPSII/Y6FU dataset, we examined growth metrics up to 6 years of age in infants fed cow milk- versus soy-based infant formulas for the first 3 months of life. Soy-fed constituted 7.33% of the subjects (*N* = 36 soy-based formula; *N* = 455 cow milk formula). There were statistically significant differences as a function of age, diet, and sex for both weight and length by mixed-effects model analyses, but the only significant diet finding between matched sex and age cohorts by multiple comparison analyses was a 13.7% decrease in body weight in females fed soy-based formula versus those fed cow milk formula, at 6 years of age, which correlated with decreased BMI (Figure 3). At 12 months of age, average female BMIs were 16.9 (breast milk), 17.7 (cow milk formula) and 19.2 (soy-based formula) and male BMIs were 17.5 (breast milk), 18.4 (cow milk formula) and 20.3 (soy-based formula). Although not statistically significant with the small number of soy subjects, the BMIs at 12 months of age are trending higher with soy-based infant formula and exceed 20 in the males (Appendix A). A healthy infant’s BMI should not be above 20, which is a risk factor for the development of obesity. Average female BMIs at 6 years of age were 16.9 (cow milk formula) and 15.1 (soy-based formula), corresponding to CDC percentiles of 83% and 48%, respectively, which are both in the normal range (Appendix A). Infants are classified as over-weight if they have a BMI ≥ 85th percentile. The average male BMIs at 6 years of age were 16.3 (cow milk formula) and 15.6 (soy-based infant formula) with CDC percentiles of 69% and 53%, respectively. Longitudinal analyses are needed to elucidate the temporal relationships between soy exposure during infancy and the development of chronic disease. It will be interesting to examine adolescent and young adult populations as a function of infant diet if further follow-up studies are conducted by the CDC. The small number of subjects in the soy cohort and possible errors in parent-reported body weights are limitations to the analysis.

### 3.9. Soy-Based Infant Formula Is Associated with an Increased Need for Support in School

Our prior work indicated increased autistic behaviors in autism and FXS with consumption of soy-based infant formula [19,20,21]. The overall prevalence of autism or developmental delay in the IFSII/Y6FU dataset was 2.86% (0.85% for females and 5.0% for males) compared to 1.85% autism (0.069% for females and 2.97% for males) reported by the Autism and Developmental Disabilities Monitoring (ADDM) Network [33]. The higher prevalence in the survey data agrees with typically higher reported values from household survey data asking a version of the question, “Has a doctor or health professional ever told you that your child had autism, Asperger’s disorder, pervasive developmental disorder, or autism spectrum disorder?”, compared to the reported prevalence by the ADDM [34]. IFPSII/Y6FU subjects reporting use of soy-based infant formula for the first 3 months of life and Y6FU outcomes trended for higher autism (*N* = 36, 5.56% autism) compared to exclusive use of breast milk (*N* = 419, 1.91% autism), and cow milk formula (*N* = 455, 3.52% autism), but was not statistically significant (*p* = 0.22) (Table 4). Binning the data by sex indicated 9.52% autism in soy-fed males (*N* = 21), but was also not statistically significant with the small number of subjects. There was a highly statistically significant, greater than 5-fold increase in needed classroom support in subjects fed soy-based infant formulas compared to both breast milk and cow milk formula, as well as a 3.1-fold increase in help needed in school comparing breast milk and soy-based infant formulas. Binning by sex, females fed soy-based infant formula exhibited higher hay fever/respiratory allergies, and males needed more help in school and support in the classroom (Appendix A).

## 4. Discussion

A minimal number of published studies have assessed the effects of direct ingestion of soy-based infant formula on laboratory animals [35,36,37,38]. Herein, we find increased body weight, altered behavior, and differential biomarker expression in WT and/or *Fmr1^KO^* mice fed SIF compared to CIF. These mouse studies corroborate recent retrospective medical record and survey analyses of human subjects indicating associations between consumption of soy-based infant formula and increased seizures, autism, gastrointestinal problems, and allergies in autism and FXS, while long-term consumption of breastmilk is protective against autism in FXS [18,19,20,21,39]. The mouse work also corroborates a national population level study based on health screening examinations linked with insurance claims data, indicating associations between consumption of soy-based infant formula, and an increased incidence of epilepsy in children and ADHD in boys [40,41].

Breastfeeding is the best nutrition for infant growth and development [42]; however, while 83% of mothers initiate breastfeeding, only 50% are breastfeeding at 6 months and 24% at 12 months [43]. Over half of newborns receive formula while in the hospital [43]. The leading infant formulas currently in use are cow milk-based, followed by soy-based. Estimates indicate that 12–36% of infants in the U.S. receive some soy-based formula during their first year of life, but there is no data regarding how many are exclusively fed soy-based infant formula [44,45,46,47]. Most infant feeding studies have not differentiated between formula types when comparing developmental milestones with breast milk feeding. Thus, despite the prevalent use of soy-based infant formulas, there have been few longitudinal studies examining their effects on health, and none in developmental disability models [30,48,49,50,51,52], which may be more prone to both positive and adverse effects in response to environmental and dietary factors.

Herein, data from the IFPSII/Y6FU study indicated significant associations between increased need for help in school and support in the classroom in response to soy-based infant formula, suggesting effects on neurological development. These data concur with a study in boys 8 years of age fed soy-based infant formula and exhibiting lower Behavior Rating Inventory of Executive Function test scores compared to those fed cow milk-based formula [53]. We did not find increased obesity in typically developing children at 6 years of age. It is interesting that the anthropometric data at 12 months of age indicates trends for increased BMI in both the male and female soy cohorts, but by 6 years of age, the soy cohorts have lower BMIs. The results at 12 months of age appear to contradict the Arkansas Beginning Study; however, a closer examination of their data, which indicates statistically significant differences between breast milk and cow milk- and soy-based infant formulas, also shows trends for increased weight and body length in the soy-based infant formula cohort at 12 months of age compared to the cow milk-based formula cohort [49]. Likewise, composite, verbal, and nonverbal intelligence indices show trends for decreasing scores with breast milk > cow-based milk formula > soy-based infant formula [51]. Our results indicate that increased body weight in response to SIF was significant in *Fmr1^KO^* but not in WT animals, suggesting that soy may have more pronounced effects in FXS and other developmental disabilities, which were excluded from the Beginnings Study. Altered growth trajectories in response to soy-based infant formula in both typically developing and neurodevelopmental populations require further investigation. Analysis of the National Health and Nutrition Examination Survey (NHANES) 2015–2016 dataset indicates that soy-based infant formula is associated with lower weight-for-length, head circumference-for-age and abdominal-for-age Z-scores among infants < 6 months of age [54].

Bench-to-bedside plans for FXS are severely limited by the lack of validated outcome measures [55]. Several candidate biomarkers are at the early stage of validation [56]. Here, we identify leptin and neprilysin as potential blood-based biomarkers that are responsive to infant diet. Leptin is a 16 kDa peptide hormone that can cross the blood-brain barrier and regulate food intake and energy expenditure via hypothalamic circuits [57]. The plasma leptin levels are highly correlated with BMI [58,59]. There are reports of both reduced and elevated leptin levels in response to phytoestrogen-enriched diets in rodents [60,61]. Leboucher and colleagues determined that leptin levels were lower in *Fmr1^KO^* mice. Their mice were fed Mucedola 4RF25 chow, which has the main ingredients of soybean meal extracted toasted, wheat, maize, and fish meal; blood samples were collected after fasting [62]. We did not find a statistically significant decrease in non-fasting leptin levels as a function of the *Fmr1^KO^* genotype, albeit there was a significant increase in leptin levels in response to SIF selectively in WT mice. Others find no statistically significant differences in leptin, cholesterol, triglycerides, adiponectin, free fatty acids, or glycerol between WT and *Fmr1^KO^* mice maintained on a Harlan Teklad low phytoestrogen laboratory chow [63]. Leptin levels are significantly higher in FXS subjects [64].

Leptin decreases the expression of neprilysin (NEP) [65]. NEP cleaves at the amino-terminal side of hydrophobic residues, thereby inactivating numerous bioactive peptides, including the insulin chain and β-amyloid (Aβ) [66]. Plasma NEP levels are associated with metabolic syndrome in human subjects and obesity in mice [67,68]. Herein, the NEP was differentially expressed as a function of SIF in both WT and *Fmr1^KO^* mice. Thus, the NEP appears to be a promising target for future investigation that could link metabolic and neurological phenotypes characteristic of FXS (Figure 4).

The strengths of the prospective mouse study include the use of littermate controls, matched diets, and complementary physiological, behavioral, and biomarker studies. The limitation of the mouse work is that metabolism varies between rodents and humans, which may hinder the translatability of the results between species. Regardless, the findings elicit significant questions regarding the potential effects of postnatal diet on neurodevelopment. Mice are the preeminent model for drug testing prior to clinical trials, and diet-drug interactions could explain why many drugs do not translate well in clinical trials. The strengths of the IFPSII dataset include its prospective design, the extensive testing of survey questions, the detail of the data collected about infants’ feeding patterns, the frequency with which data were collected, the inclusion of questions that address most topic areas likely to affect infant feeding, the large national sample size, and the 6-year follow-up study. The strengths of the Y6FU dataset include that it is the largest longitudinal study in the U.S. to examine the long-term consequences of infant feeding with 52.1% of the IFPSII cohort completing the follow-up study. The limitations include the sample demographic, which, although well-distributed throughout the U.S., is not representative of the U.S. population; all data are self-reported by the mother; no medical records were examined to confirm infant health, weight, length, or any other characteristic; lack of consideration of complementary foods; only healthy infants were included. Infants with special needs or medical problems would be predicted to have increased feeding problems and to be more susceptible to diets or dietary components that exacerbate seizures and other neurological phenotypes.

## 5. Conclusions

In summary, single-source, soy-based diets are associated with weight gain and differential neurological outcomes in a mouse model of FXS. The findings are important because there is a paucity of studies regarding the effects of pediatric nutrition on neurological development and metabolism, particularly regarding infants with developmental disabilities that are comorbid with obesity, including ASD [81] and FXS [16]. This study could have powerful translational implications in terms of identifying a dietary intervention (restriction of soy-based infant formula) to improve medical outcomes in vulnerable populations. Considering the projected prevalence and cost of ASD [34], the role of early-life diet on neurological development warrants further investigation. In the case of FXS, newborn screening would be required to identify infants. Parents typically recognize developmental delays in their children with FXS as early as 9 months of age, but the average age of diagnosis is 36 months [82]. The major challenge confronting the implementation of newborn screening for FXS is the lack of an effective early-life treatment [82].

Future directions include a rigorous assessment of the effects of exactly matched casein- and soy protein-based diets on weight gain and behavior in WT and *Fmr1^KO^* mice. The CIF and SIF diets utilized here were formulated from infant formulas optimized for human nutrition and thus contained significantly less protein and more fat than typical mouse diets. Moving forward, we will assess the effects of soy protein as a function of FMRP status with exactly matched casein- and soy-protein-based diets formulated for optimal mouse nutrition. Herein, fast-growing juvenile animals were more susceptible to the effect of diet on weight gain, but it will be important to assess dietary effects throughout the lifespan and as a function of developmental disability and drug treatment. For example, soy greatly reduces the plasma concentration of valproic acid [83]. Thus, dietary components that affect the pharmacokinetics and pharmacodynamics of drugs could underlie issues with translating promising preclinical FXS therapeutics into humans. Unfortunately, most publications on FXS rodent studies do not report diet details. It will also be important to perform studies to validate the model proposed in Figure 4. NEP appears to be a promising biomarker that could link metabolic and neurological phenotypes in FXS and deserves further investigation. FDA approved drugs have been identified that act as potential NEP inhibitors [84].

## Figures and Tables

**Figure 1 cells-11-01350-f001:**
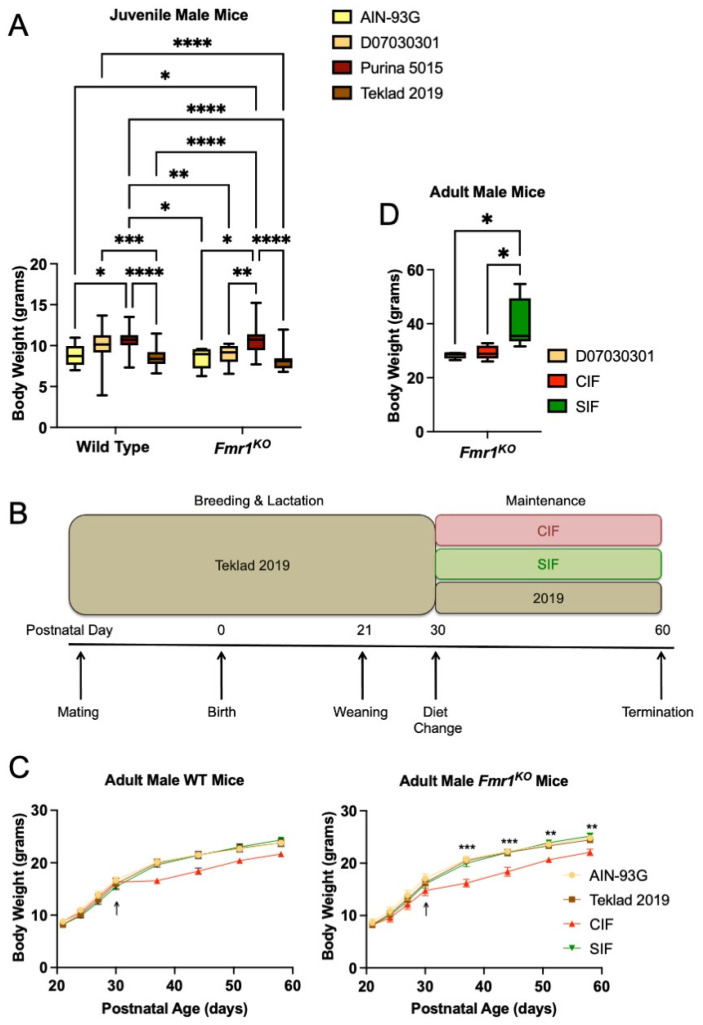
Mouse body weight as a function of *Fmr1* genotype and diet. (**A**) Body weight of juvenile WT and *Fmr1^KO^* male mice in response to rodent diets. Cohorts include WT/AIN-93G (*n* = 10), WT/D07030301 (*n* = 80), WT/Purina 5015 (*n* = 29), WT/Teklad 2019 (*n* = 29), *Fmr1^KO^*/AIN-93G (*n* = 7), *Fmr1^KO^*/D07030301 (*n* = 17), *Fmr1^KO^*/Purina 5015 (*n* = 32) and *Fmr1^KO^*/Teklad 2019 (*n* = 23). Mice were maintained on the indicated diets throughout gestation and lactation and weighed on postnatal day 21 (P21) immediately prior to seizure testing and average weight in grams with SEM (*y*-axis) plotted against genotype (*x*-axis). Statistics were determined by 2-way ANOVA and Tukey’s multiple comparison tests denoted by *p* < 0.05 (*), *p* < 0.01 (**), *p* < 0.001 (***), and *p* < 0.0001 (****). Two ANOVA results include interaction F(3219) = 1.497, *p* = 0.22; genotype F(1219) = 2.544, *p* = 0.11; diet F(3219) = 23.68, *p* < 0.0001. (**B**) Schematic of experimental plan. (**B**) The experimental plan includes *Fmr1^HET^* females were bred and maintained on Teklad2019 (grain-based but low phytoestrogen levels) diet throughout gestation and lactation. Male WT and *Fmr1^KO^* pups were weaned at P21 and randomized to diets (Teklad 2019, CIF, SIF) at P30. The pellet infant formula diets were prepared from Enfamil Premium (cow milk-based, CIF) versus Enfamil ProsSobee (soy protein-based, SIF) infant formulas (powdered formula purchased from Walmart and pellets synthesized by Research Diets, Inc.). The mice were weighed at P21, P24, P27, and P30 prior to the diet change, and once per week for 4 weeks after the diet change. A second reference diet included AIN-93G in which pups were bred and maintained on AIN-93G throughout the study. (**C**) Body weight of adolescent WT and *Fmr1^KO^* male mice in response to cow milk- and soy-based infant formula diets for 1 month. WT cohorts include AIN-93G (*n* = 9), Teklad 2019 (*n* = 6), CIF (*n* = 13), and SIF (*n* = 10). Statistics were determined by 2-way repeated measures ANOVA: age x diet F(21,238) = 9.487, *p* < 0.0001; age F(7238) = 1163, *p* < 0.0001; diet F(3,34) = 1.994, *p* = 0.13; subject F(34,238) = 20.60, *p* < 0.0001). *Fmr1^KO^* cohorts include AIN-93G (*n* = 5), Teklad 2019 (*n* = 10), CIF (*n* = 7) and SIF (*n* = 6). Statistics were determined by 2-way repeated measures ANOVA: age x diet F(21,168) = 6.319, *p* < 0.0001; age F(7168) = 1463, *p* < 0.0001; diet F(3,24) = 3.726, *p* = 0.0249; subject F(24,168) = 28.47, *p* < 0.0001). Statistics denoted by *p* < 0.01 (**) and *p* < 0.001 (***) for CIF versus SIF. The *Fmr1^KO^* CIF cohort is statistically different from the *Fmr1^KO^* AIN-93G and Teklad 2019 cohorts at the same time points (statistics not shown). (**D**) Body weight of adult male *Fmr1^KO^* mice in response to infant formula diets. Adult male *Fmr1^KO^* mice were randomized to D07030301, CIF or SIF diets at 2–3 months of age for breeding. Mice in the SIF cohort (*n* = 5) became noticeably obese compared to those fed CIF (*n* = 5) or D07030301 (*n* = 4) and thus all the mice were weighed at approximately 4 months of age. Body weight is plotted versus diet in a box-whisker plot. Statistics were determined by one-way ANOVA and Tukey’s multiple comparison tests: F(2,11) = 6.000, *p* = 0.017, and denoted by *p* < 0.05 (*).

**Figure 2 cells-11-01350-f002:**
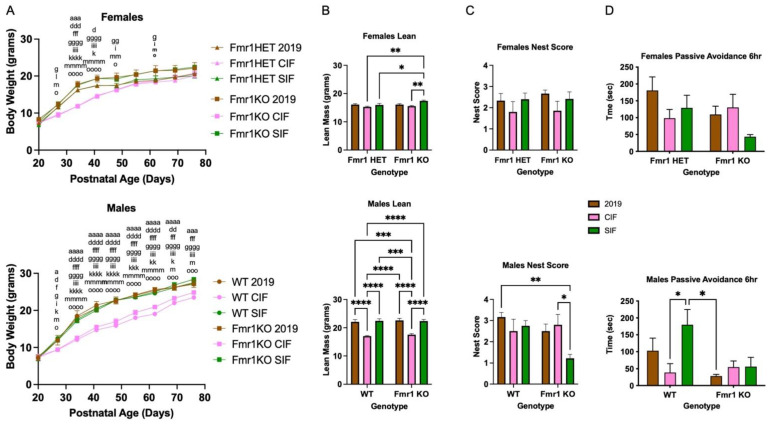
Body weight and behavior of adolescent WT and *Fmr1^KO^* male mice in response to cow milk- and soy-based infant formula diets after 2 months treatment. *Fmr1^HET^* females were bred and maintained on Teklad2019 (grain-based but low phytoestrogen levels) diet throughout gestation and lactation. Female *Fmr1^HET^* and *Fmr1^KO^* and male WT and *Fmr1^KO^* pups were weaned at P20 and randomized to diets (Teklad 2019, CIF, SIF). (**A**) The mice were weighed once per week for 2 months. Female cohorts included *Fmr1^HET^*/Teklad 2019 (*n* = 4), *Fmr1^HET^*/CIF (*n* = 5), *Fmr1^HET^*/SIF (*n* = 6), *Fmr1^KO^*/Teklad 2019 (*n* = 7), *Fmr1^KO^*/CIF (*n* = 7) and *Fmr1^KO^*/SIF (*n* = 6). Statistics were determined by 2-way mixed effects model ANOVA: time F(8230) = 1413, *p* < 0.0001; genotype/diet F(5,29) = 5.122, *p* = 0.0017; time x genotype/diet F(40,230) = 8.714, *p* < 0.0001. Male cohorts included the following: WT/Teklad 2019 (*n* = 6), WT/CIF (*n* = 6), WT//SIF (*n* = 4), *Fmr1^KO^*/Teklad 2019 (*n* = 5), *Fmr1^KO^*/CIF (*n* = 5) and *Fmr1^KO^*/SIF (*n* = 7). Statistics were determined by 2-way mixed-effects model ANOVA: time F(8216) = 2550, *p* < 0.0001; genotype/diet F(5,27) = 16.78, *p* < 0.0001; time x genotype/diet F(40,216) = 13.34, *p* < 0.0001. Tukey’s multiple comparison tests are denoted by *p* < 0.05 (^x^), *p* < 0.01 (^xx^), *p* < 0.001 (^xxx^), and *p* < 0.0001 (^xxxx^) where females x = a for *Fmr1^HET^* Teklad 2019 versus *Fmr1^HET^* CIF, x = b for *Fmr1^HET^* Teklad 2019 versus *Fmr1^HET^* SIF, x = c for *Fmr1^HET^* Teklad 2019 versus *Fmr1^KO^* 2019, x = d for *Fmr1^HET^* Teklad 2019 versus *Fmr1^KO^* CIF, x = e for *Fmr1^HET^* Teklad 2019 versus *Fmr1^KO^* SIF, x = f for *Fmr1^HET^* CIF versus *Fmr1^HET^* SIF, x = g for *Fmr1^HET^* CIF versus *Fmr1^KO^* 2019, x = h for *Fmr1^HET^* CIF versus *Fmr1^KO^* CIF, x = i for *Fmr1^HET^* CIF versus *Fmr1^KO^* SIF, x = j for *Fmr1^HET^* SIF versus *Fmr1^KO^* 2019, x = k for *Fmr1^HET^* SIF versus *Fmr1^KO^* CIF, x = l for *Fmr1^HET^* SIF versus *Fmr1^KO^* SIF, x = m for *Fmr1^KO^* Teklad 2019 versus *Fmr1^KO^* CIF, x = n for *Fmr1^KO^* Teklad 2019 versus *Fmr1^KO^* SIF, and x = o for *Fmr1^KO^* CIF versus *Fmr1^KO^* SIF, and males x = a for WT Teklad 2019 versus WT CIF, x = b for WT Teklad 2019 versus WT SIF, x = c for WT Teklad 2019 versus *Fmr1^KO^* 2019, x = d for WT Teklad 2019 versus *Fmr1^KO^* CIF, x = e for WT Teklad 2019 versus *Fmr1^KO^* SIF, x = f for WT CIF versus WT SIF, x = g for WT CIF versus *Fmr1^KO^* 2019, x = h for WT CIF versus *Fmr1^KO^* CIF, x = i for WT CIF versus *Fmr1^KO^* SIF, x = j for WT SIF versus *Fmr1^KO^* 2019, x = k for WT SIF versus *Fmr1^KO^* CIF, x = l for WT SIF versus *Fmr1^KO^* SIF, x = m for *Fmr1^KO^* Teklad 2019 versus *Fmr1^KO^* CIF, x = n for *Fmr1^KO^* Teklad 2019 versus *Fmr1^KO^* SIF, and x = o for *Fmr1^KO^* CIF versus *Fmr1^KO^* SIF. (**B**) EchoMRI measurement of lean body mass. Female cohorts include *Fmr1^HET^*/Teklad 2019 (*n* = 4), *Fmr1^HET^*/CIF (*n* = 5), *Fmr1^HET^*/SIF (*n* = 6), *Fmr1^KO^*/Teklad 2019 (*n* = 6), *Fmr1^KO^*/CIF (*n* = 7) and *Fmr1^KO^*/SIF (*n* = 6). Statistics were determined by two-way ANOVA: interaction F(2,28) = 2.702, *p* = 0.085; genotype F(1,28) = 4.410, *p* = 0.045; diet F(2,28) = 7.438, *p* = 0.0026. Male cohorts include WT/Teklad 2019 (*n* = 6), WT/CIF (*n* = 6), WT/SIF (*n* = 4), *Fmr1^KO^*/Teklad 2019 (*n* = 6), *Fmr1^KO^*/CIF (*n* = 5) and *Fmr1^KO^*/SIF (*n* = 7). Statistics were determined by two-way ANOVA: interaction F(2,28) = 0.1425, *p* = 0.87; genotype F(1,28) = 0.3857, *p* = 0.54; diet F(2,28) = 46.03, *p* < 0.0001. Tukey’s multiple comparison tests denoted by *p* < 0.05 (*), *p* < 0.01 (**), *p* < 0.001 (***), and *p* < 0.0001 (****). Total body weight, fat mass, total water mass, and free water mass measurements assessed by EchoMRI are provided in the Appendix A. (**C**) Nest building female cohorts include *Fmr1^HET^*/Teklad 2019 (*n* = 3), *Fmr1^HET^*/CIF (*n* = 5), *Fmr1^HET^*/SIF (*n* = 5), *Fmr1^KO^*/Teklad 2019 (*n* = 6), *Fmr1^KO^*/CIF (*n* = 7) and *Fmr1^KO^*/SIF (*n* = 6). Statistics were determined by two-way ANOVA: interaction F(2,26) = 0.008736, *p* = 0.92; genotype F(1,26) = 0.1821, *p* = 0.67; diet F(2,26) = 1.86, *p* = 0.18. Male cohorts include WT/Teklad 2019 (*n* = 6), WT/CIF (*n* = 6), WT/SIF (*n* = 4), *Fmr1^KO^*/Teklad 2019 (*n* = 6), *Fmr1^KO^*/CIF (*n* = 5) and *Fmr1^KO^*/SIF (*n* = 7). Statistics were determined by two-way ANOVA: interaction F(2,28) = 2.94, *p* = 0.069; genotype F(1,28) = 4.376, *p* = 0.046; diet F(2,28) = 2.845, *p* = 0.075. Tukey’s multiple comparison tests denoted by *p* < 0.05 (*) and *p* < 0.01 (**). (**D**) Passive avoidance female cohorts include *Fmr1^HET^*/Teklad 2019 (*n* = 4), *Fmr1^HET^*/CIF (*n* = 5), *Fmr1^HET^*/SIF (*n* = 6), *Fmr1^KO^*/Teklad 2019 (*n* = 6), *Fmr1^KO^*/CIF (*n* = 7) and *Fmr1^KO^*/SIF (*n* = 6). Statistics were determined by two-way ANOVA: interaction F(2,28) = 2.093, *p* = 0.14; genotype F(1,28) = 2.530, *p* = 0.12; diet F(2,28) = 1.642, *p* = 0.21. Male cohorts include WT/Teklad 2019 (*n* = 6), WT/CIF (*n* = 6), WT/SIF (*n* = 4), *Fmr1^KO^*/Teklad 2019 (*n* = 6), *Fmr1^KO^*/CIF (*n* = 5) and *Fmr1^KO^*/SIF (*n* = 7). Statistics were determined by two-way ANOVA: interaction F(2,28) = 2.998, *p* = 0.066; genotype F(1,28) = 6.844, *p* = 0.014; diet F(2,28) = 3.229, *p* = 0.055. Tukey’s multiple comparison tests denoted by *p* < 0.05 (*). Data shown for 6-hours post-foot shock. Results were not statistically different for 24- and 48-h post-foot shock.

**Figure 3 cells-11-01350-f003:**
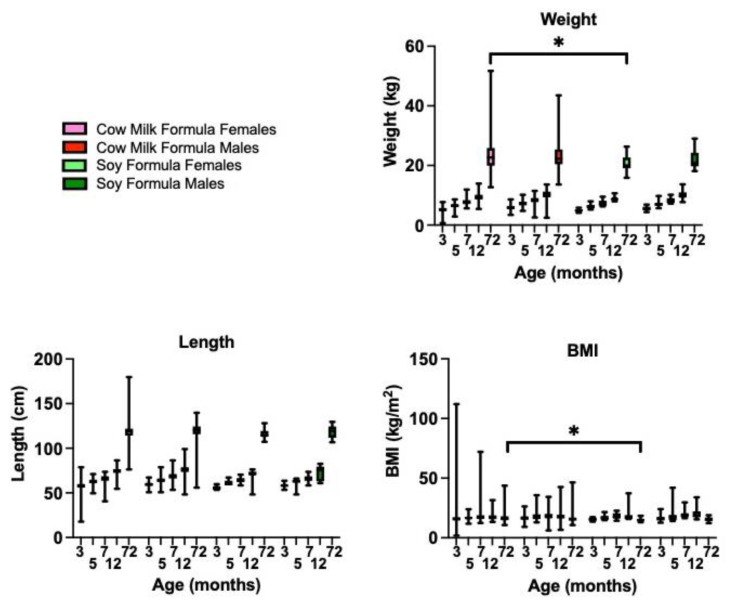
Infant feeding during the first 3 months of life is associated with growth metrics and adverse outcomes in children at 6 years of age. Growth cohorts included females fed cow milk formula (*n* = 240), males fed cow milk formula (*n* = 215), females fed soy formula (*n* = 15), and males fed soy formula (*n* = 21). Statistics were determined by two-way ANOVA with mixed-effects analysis for body weight (age *p* < 0.0001, diet *p* = 0.0124, sex *p* = 0.0049, age x diet *p* = 0.0010, age x sex *p* = 0.9733, diet x sex *p* = 0.3370, and age x diet x sex *p* = 0.4498); body length (age *p* < 0.0001, diet *p* = 0.0134, sex *P* = 0.0059, age x diet *p* = 0.8580, age x sex *p* = 0.9953, diet x sex *p* = 0.7330, and age x diet x sex *p* = 0.9936); BMI (age *p* < 0.0001, diet *p* = 0.7892, sex *p* = 0.1773, age x diet *p* = 0.0861, age x sex *p* = 0.7547, diet x sex *p* = 0.4810, and age x diet x sex *p* = 0.9901). Statistical significance for age-matched comparisons is denoted by *p* < 0.05 (*).

**Figure 4 cells-11-01350-f004:**
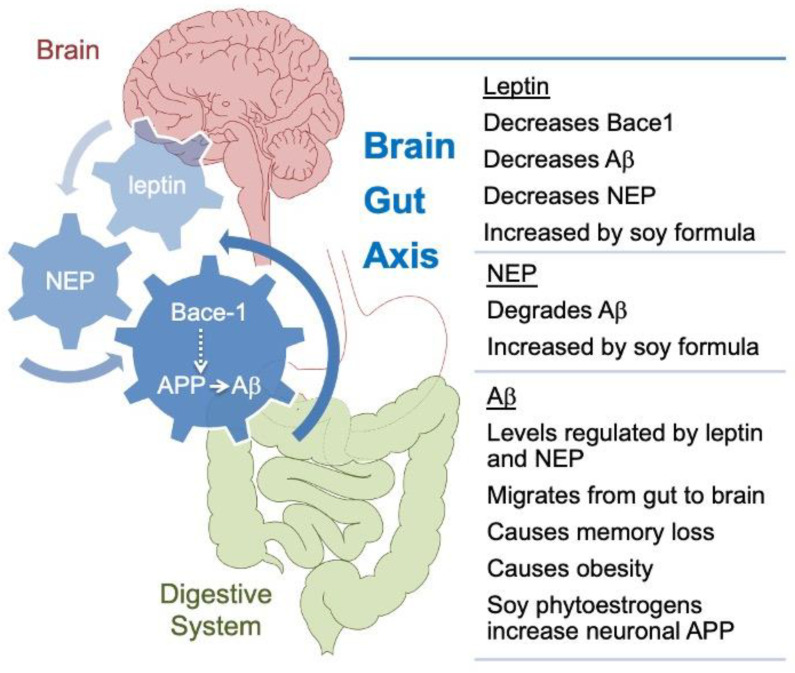
Model of proposed gut-brain interactions underlying soy effects in *Fmr1^KO^* mice. SIF could contribute to increased weight gain through a pathway involving APP metabolism, leptin, and NEP. It has been hypothesized that amyloidogenic processing of APP in peripheral tissues plays a key role in the response to nutrient excess and that this could contribute to the pathogenesis of metabolic diseases [69]. Aβ, which is generated by β-secretase 1 (Bace1) and γ-secretase processing of APP, can affect numerous metabolic processes [70,71]. APP and Aβ levels are dysregulated in mouse and human models of FXS [72,73], a disorder comorbid with increased BMI [74] and where more severe phenotypes are associated with consumption of single-source soy-based diets [17,20,21]. It is known that soy phytoestrogens increase APP synthesis in primary cultured mouse neuronal cells [17], which would provide more template for Bace-1 processing, but the mechanism(s) mediating the effects of dietary soy are not known [75]. Leptin and NEP were identified as plasma-based biomarkers responsive to SIF. These biomarkers are linked with APP processing [76], obesity [67,77,78], and each other [65]. Aβ can migrate from the gastrointestinal (GI) tract to the brain and cause cognitive impairment, but soy flavonoids have a protective effect [79]. Leptin attenuates the detrimental effects of Aβ on spatial memory in rats [80]; herein, SIF increased leptin levels and improved 6-hr recall in the passive avoidance test in WT mice. Soy-based diets negatively impact seizure and autism outcomes in mouse and human models [17,18,19,20,21] consistent with the increased need for classroom support and help needed in school. In total, these data support a model of overlapping feedback loops with U-shaped response curves, involving APP processing, leptin, and neprilysin that underlies diet-induced metabolic and neurological outcomes in response to consumption of single-source soy-based diets.

**Table 1 cells-11-01350-t001:** Differentially expressed plasma proteins in WT mice dependent on diet.

Protein Target	Fold Change	Post Hoc *t*-Test (*p*)
6Ckine	0.45 ^1^	≤0.05
Axl	2.02 ^1^	≤0.02
	1.76 ^2^	≤0.05
B7-1	1.48 ^1^	≤0.05
	2.04 ^2^	≤0.00
	0.73 ^3^	≤0.05
CD30	1.35 ^1^	≤0.02
CD36	2.40 ^1^	≤0.00
	2.37 ^3^	≤0.05
Epiregulin	1.54 ^1^	≤0.05
Galactin-7	0.44 ^1^	≤0.04
ICAM-1	6.08 ^1^	≤0.04
	3.68 ^3^	≤0.05
IL-1 R4	0.12 ^2^	≤0.01
IL-2 Ra	1.43 ^1^	≤0.00
Leptin	5.68 ^2^	≤0.02
	0.11 ^3^	≤0.02
MFG-E8	4.34 ^3^	≤0.00
MMP-10	4.23 ^1^	≤0.01
	2.07 ^2^	≤0.00
	2.04 ^3^	≤0.04
Neprilysin	4.38 ^2^	≤0.03
	0.25 ^3^	≤0.05

^1^ CIF versus 2019. ^2^ SIF versus 2019. ^3^ CIF versus SIF.

**Table 2 cells-11-01350-t002:** Differentially expressed plasma proteins in *Fmr1^KO^* mice dependent on diet.

Protein Target	Fold Change	Post Hoc *t*-Test (*p*)
B7-1	2.18 ^2^	≤0.01
CXCL 16	0.77 ^2^	≤0.04
Epiregulin	1.56 ^1^	≤0.04
IGFBP-5	0.63 ^1^	≤0.04
	0.73 ^3^	≤0.05
IL-1 R4	0.17 ^1^	≤0.02
Lipocalin-2	8.73 ^2^	≤0.01
Neprilysin	2.45 ^2^	≤0.05
P-Cadherin	2.11 ^2^	≤0.00
Prolactin	3.60 ^1^	≤0.00
Renin 1	2.28 ^1^	≤0.03
VEGF-B	4.42 ^1^	≤0.05

^1^ CIF versus 2019. ^2^ SIF versus 2019. ^3^ CIF versus SIF.

**Table 3 cells-11-01350-t003:** Differentially expressed plasma proteins dependent on genotype.

Protein Target	Fold Change	Post Hoc *t*-Test (*p*)
6Ckine	3.27 ^1^	≤0.02
B7-1	1.87 ^1^	≤0.02
	1.75 ^3^	≤0.01
BLC	2.08 ^1^	≤0.03
Galactin-7	2.96 ^1^	≤0.00
ICAM-1	3.71 ^2^	≤0.05
MFG-E8	4.77 ^2^	≤0.00
Prolactin	0.34 ^2^	≤0.02
TRANCE	2.58 ^1^	≤0.02
VEGF-B	4.12 ^1^	≤0.02

^1^ 2019, WT versus *Fmr1^KO^*. ^2^ CIF, WT versus *Fmr1^KO^*. ^3^ SIF, WT versus *Fmr1^KO^*.

**Table 4 cells-11-01350-t004:** Incidence of adverse outcomes in IFPSII/Y6FU study population as a function of infant diet.

Metric	Breast	Cow Milk Formula	Soy Formula	*p* ^1^
N	419	455	36	
IEP	10%	15%	17%	0.093
Speech Therapy	9.3%	15%	17%	0.037 ^2^
Occupational Therapy	3.3%	4.8%	8.3%	0.26
Help in School	5.5%	11%	17%	0.0035 ^3^
Support in Classroom	2.6%	2.2%	14%	0.0003 ^4^
Hay Fever or Respiratory Allergy	18%	25%	31%	0.024 ^5^
Asthma	8.1%	13%	14%	0.07
ADD or ADHD	1.2%	5.1%	0%	0.0024 ^6^
Autism or Developmental Delay	1.9%	3.5%	5.6%	0.22

^1^ Chi square 3 × 2. ^2^ *p* = 0.014 breast versus cow milk formula, *p* = 0.16 breast versus soy formula, *p* = 0.75 cow milk formula versus soy formula by Chi square 2 × 2. ^3^ *p* = 0.0033 breast versus cow milk formula, *p* = 0.0084 breast versus soy formula, *p* = 0.3 cow milk formula versus soy formula by Chi square 2.2. ^4^ *p* = 0.68 breast versus cow milk formula, *p* = 0.0004 breast versus soy formula, *p* < 0.0001 cow milk versus soy formula by Chi square 2 × 2. ^5^ *p* = 0.016 breast versus cow milk formula, *p* = 0.063 breast versus soy formula, *p* = 0.43 cow milk formula versus soy formula by Chi square 2 × 2. ^6^ *p* = 0.0012 breast versus cow milk formula by Chi square 2 × 2.

## Data Availability

The mouse data presented in this study are available in the article and Appendix A. The CDC provided copies of the IFPSII and follow-up human study datasets.

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
