# Peer review of "Effects of Soy-Based Infant Formula on Weight Gain and Neurodevelopment in an Autism Mouse Model"

_cells, 2022, doi:10.3390/cells11081350_

Round 1
Reviewer 1 Report
This study aimed to analyze the effect of diet on body weight, and to assess plasma protein biomarker expression and behavior in response to diet. They found that juvenile Fmr1KO mice fed a soy protein-based rodent chow throughout gestation and postnatal development exhibit increased weight
gain compared to mice fed casein-based purified ingredient diet or grain-based, low phytoestrogen chow, and adolescent and adult Fmr1KO mice fed a soy-based infant formula diet exhibited increased weight gain compared to reference diets. Increased body mass was due to increased lean mass. Wild type male mice fed soy-based infant formula exhibited increased learning in a passive avoidance paradigm and Fmr1KO male mice had a deficit in nest building. Thus, they concluded that consumption of soy-based diets increases weight gain and affects behavior, and soy-based infant formula diet was associated with altered expression of numerous plasma proteins including the adipose hormone leptin and the b-amyloid degrading enzyme neprilysin, indicating that single source, soy-based diets may contribute to the development of obesity and the exacerbation of neurological phenotypes in developmental disabilities such as FXS.Strengths:The experimental design is completed and rigorous, and the data is presented clearly. The statistical analysis is good and test methods were chosen correctly.
Weakness:
- If the soy-based diet contribute the weight gain and behavioral deficits in the FXS, is the behavioral deficits and increased weight gain able to be rescued if the soy-based diet is replaced by casein-based diet in the early adult stage?
- The soy-based diet is able to affect the body weight, does it affect the locomotor activity of the fmr1 ko mice? Locomotor activity is important factor for the other behaviors.
Author Response
Review 1: This study aimed to analyze the effect of diet on body weight, and to assess plasma protein biomarker expression and behavior in response to diet. They found that juvenile Fmr1KO mice fed a soy protein-based rodent chow throughout gestation and postnatal development exhibit increased weight gain compared to mice fed casein-based purified ingredient diet or grain-based, low phytoestrogen chow, and adolescent and adult Fmr1KO mice fed a soy-based infant formula diet exhibited increased weight gain compared to reference diets. Increased body mass was due to increased lean mass. Wild type male mice fed soy-based infant formula exhibited increased learning in a passive avoidance paradigm and Fmr1KO male mice had a deficit in nest building. Thus, they concluded that consumption of soy-based diets increases weight gain and affects behavior, and soy-based infant formula diet was associated with altered expression of numerous plasma proteins including the adipose hormone leptin and the b-amyloid degrading enzyme neprilysin, indicating that single source, soy-based diets may contribute to the development of obesity and the exacerbation of neurological phenotypes in developmental disabilities such as FXS.
Strengths:
The experimental design is completed and rigorous, and the data is presented clearly. The statistical analysis is good and test methods were chosen correctly. Thank you.
Weakness:
- If the soy-based diet contribute the weight gain and behavioral deficits in the FXS, is the behavioral deficits and increased weight gain able to be rescued if the soy-based diet is replaced by casein-based diet in the early adult stage? This is a good question that we have not addressed yet. We have added this to the new Future Directions section at the end of the paper.
- The soy-based diet is able to affect the body weight, does it affect the locomotor activity of the fmr1 ko mice? Locomotor activity is important factor for the other behaviors. This is also a good question that we are currently investigating. We have randomized WT and Fmr1KO animals to exactly matched soy and casein-based diets optimized for rodent health and performed a behavioral battery that includes rotarod, actigraphy and passive avoidance. We plan to have the results ready for publication later this year. This has also been added to the Future Directions.
Reviewer 2 Report
This is an interesting manuscript that raises an important issue of the influence of soy-based diet on neurodevelopment and weight gain in fragile X syndrome. The experiments were well-planned and the analyses appear to be carefully conducted. The study brings novelty and the paper is worth publication. I cannot find any weak points. Some future perspectives can be added in the Conclusion section. What should be the next steps in studies on this matter? I have also one minor technical comment – figure legends should be put below the figures.
Author Response
Review 2: This is an interesting manuscript that raises an important issue of the influence of soy-based diet on neurodevelopment and weight gain in fragile X syndrome. The experiments were well-planned and the analyses appear to be carefully conducted. The study brings novelty and the paper is worth publication. I cannot find any weak points. Some future perspectives can be added in the Conclusion section. What should be the next steps in studies on this matter? I have also one minor technical comment – figure legends should be put below the figures. Thank you. A Future Directions section has been added to the Conclusion section. Figure legends have been moved below the figures.